# Root Canal Configuration and Its Relationship with Endodontic Technical Errors and Periapical Status in Premolar Teeth of a Saudi Sub-Population: A Cross-Sectional Observational CBCT Study

**DOI:** 10.3390/ijerph20021142

**Published:** 2023-01-09

**Authors:** Rayan Suliman Al Yahya, Mustafa Hussein Al Attas, Muhammad Qasim Javed, Kiran Imtiaz Khan, Sundus Atique, Ayman M. Abulhamael, Hammam Ahmed Bahammam

**Affiliations:** 1Department of Conservative Dental Sciences and Endodontics, College of Dentistry, Qassim University, Buraidah 52571, Saudi Arabia; 2Department of Operative Dentistry, Frontier Medical and Dental College, Abbottabad 22030, KPK, Pakistan; 3College of Dental Medicine, Qatar University, Doha 2713, Qatar; 4Department of Endodontics, Faculty of Dentistry, King Abdulaziz University, P.O. Box 80209, Jeddah 21589, Saudi Arabia; 5Department of Pediatric Dentistry, Faculty of Dentistry, King Abdulaziz University, P.O. Box 80209, Jeddah 21589, Saudi Arabia

**Keywords:** cone-beam computed tomography, endodontic errors, endodontics, oral diseases, oral health, oral radiology, root canal configuration

## Abstract

Endodontic technical errors are the foremost cause of treatment failure. A thorough understanding of root canal configuration (RCC) is essential to prevent these iatrogenic errors. This study used CBCT images to determine the association between root canal configuration, endodontic technical errors, and periapical status. CBCT images of 101 patients, including total of 212 obturated premolars (256 canals) were assessed. RCCs were classified according to the Vertucci system. The presence of endodontic errors and periapical lesions associated with each RCC was noted. Presence or absence of coronal restoration and its association with periapical radiolucency was recorded. The most frequent RCC was Type I (199 cases; 77.73%), followed by Type II (26 cases; 10.15%), Type IV (22 cases; 8.59%), Type V (4 cases; 1.56%), Type III (4 cases; 1.56%), and Type VI (1 case; 0.39%). Under-filling and non-homogeneous filling were the most common technical errors. Prevalence of periapical radiolucency was 81% in the presence of technical errors. The absence of coronal restoration caused apical lesions in 93% of cases. The frequency of endodontic technical errors increased as the root canal configurations became more complex. Periapical lesions occurred more often in teeth with endodontic errors and/or absent coronal restoration.

## 1. Introduction

Root canal treatment is a complex multi-step procedure designed to salvage a tooth with irreversibly inflamed or necrotic pulp. The clinical procedure involves gaining access to the root canal, followed by a thorough cleaning, shaping of the entire root canal system, and later filling the prepared canals with gutta-percha. Successful treatment is indicated by adequate periapical tissue healing. Endodontic technical errors can result in incomplete debridement and poor obturation of the root canal, allowing microorganisms to survive and multiply there. These microorganisms can then leak to the apex and peri-radicular tissues and cause infection, resulting in endodontic failure [1].

After molars, premolars are the teeth most frequently referred for root canal treatment [2,3]. Premolars exhibit complex anatomical variations. Numerous studies have concluded that mandibular premolars display variable root canal anatomies [4]. Typically, a mandibular premolar involves a single root containing a single canal. However, extra roots and additional canals have also been reported in the literature [5]. The frequency of mandibular premolars with two or more canals has been reported to range from 13.7% to 46% [6,7]. This diversity of morphological patterns is not limited to mandibular premolars, with maxillary premolars also displaying similar morphological variants. In fact, the maxillary second premolar is the only tooth that exhibits all eight possible canal configurations described in the Vertucci classification system [1,8]. These variations in the numbers of roots and root canals and in the pulpal configuration mean that premolars are among the most challenging teeth to treat endodontically. If not adequately assessed pre-operatively, anatomical complexities can lead to an increase in the incidence of endodontic technical errors, resulting in endodontic failure. One previous study concluded that the mandibular first premolar represents the tooth with the highest failure rate, and also reported the occurrence of multiple flare-ups while root canal treatment was being carried out [9]. Nascimento et al. conducted a study on premolar teeth and reported an association between endodontic technical errors and periapical lesions [10]. Research by Baruwa et al. into the association between untreated root canals and periapical lesions reported a high prevalence of apical periodontitis in endodontically treated teeth with missed root canals [11]. Endodontic technical errors are the foremost cause of treatment failure. To prevent iatrogenic errors, it is necessary for the practicing clinician to have comprehensive knowledge and understanding of normal and aberrant anatomy of the root canal system [5,12,13]. The numbers and configuration of roots and root canals can vary among individuals of different ages, ethnicities, and genders [14]. Awareness of complexities in the root canal system not only aids canal negotiation, instrumentation, and obturation, but also helps in avoiding errors such as over- or underfilling, missed canals, void formation, instrument separation, ledge formation, zipping, and apical transportation of the canals [8,15,16].

Several methods and aids have been developed to help clinicians visualize and assess the internal morphological patterns of teeth. Loupes, dental microscopes, tooth clearing, canal staining, and conventional 2-D radiography have traditionally been used by dental practitioners for these purposes [17]. Conventional and digital radiographs have limited value in the assessment of tooth morphology, as they produce two-dimensional images with superimposition in the buccolingual plane [18]. 

Cone-beam computed tomography is a revolutionary diagnostic tool that has proven superior to periapical radiographs for visualizing pulp canal configuration [19,20]. The features of CBCT that advocate its use for endodontic investigation include higher resolution, increased accuracy, three-dimensional imaging of the teeth, elimination of overlaps, earlier detection of apical lesions, and 3-D reconstruction of root canal systems [20,21,22,23,24,25,26]. In a joint position statement, the American Association of Endodontists (AAE) and American Academy of Oral and Maxillofacial Radiology (AAOMR) recommend the use of limited field-of-view CBCT for endodontic evaluation of teeth. The recommendations include using CBCT for preoperative assessment of complex morphology, intraoperative examination of teeth with root canal anomalies or extra canals, evaluation of teeth for non-surgical or surgical retreatment, and teeth that have experienced endodontic treatment errors [27].

Many studies have assessed and classified the anatomy of the root canal in premolar teeth, but limited research has been carried out to establish the relation between canal configuration, endodontic technical errors, and their effect on the periapical tissues. The aim of the current study was to use CBCT to classify the pulpal canal configuration of root-canal-treated premolars according to the Vertucci classification system, and to determine their association with endodontic technical errors, treatment quality, and apical status.

## 2. Materials and Methods

### 2.1. Study Outcomes, Sample Size Calculation, Research Protocol

The prevalence of different root canal configurations in root-canal-treated premolar teeth according to Vertucci’s classification (primary outcome) and the association of root canal configurations with endodontic technical errors, treatment quality, and apical status (secondary outcome) were determined in a Saudi subpopulation. The analysis was conducted by one general dental practitioner (GDP) with two years of experience and two endodontists, each with more than five years of experience. Data were obtained by evaluating pre-existing CBCT images from the archives of the oral radiology department in a college of dentistry, according to the American Association of Endodontists’ position statement [27]. The evaluators were calibrated with the aid of CBCT images and subsequently, were trained to adjust multi-plane views during evaluation in order to ensure accurate assessments. Sample size was determined by taking into account previous relevant cross-sectional research conducted in Brazil [28]. Using a G-power sample-size calculator, a total of 77 patients was calculated to be an acceptable sample size, with an effect size of 0.32, confidence level 5%, and 80% power [29]. The current research was conducted in accordance with the guidelines published for epidemiologic cross-sectional studies on root and root canal anatomy using CBCT technology [30], after acquiring approval from the Committee of Health Research Ethics, Deanship of Scientific Research, Qassim University, Saudi Arabia (Approval no:12-10-15).

### 2.2. Sample Selection, Data Acquisition, and Screening Method 

The initial study sample included CBCT images of 306 Saudi patients, and was acquired by screening for nationality in the patient files from the archives of the oral radiology department. Subsequently, the CBCT scans were screened by one general dental practitioner (GDP) with two years of experience and two endodontists each with more than five years of experience, according to the inclusion criterion of CBCT images with a minimum of one obturated premolar tooth. The criteria for exclusion were images without obturated premolar teeth, teeth with root fractures or suspicion of such, and/or teeth with endodontic–periodontal lesions. CBCT scans with discrepancies were also excluded to avoid misjudgements. The final sample comprised CBCT images of 101 patients (39 females and 62 males, mean age 43.5 ± 10.6 years), with a total of 212 obturated premolar teeth (256 canals). All CBCT scans were obtained using a Sirona Galileos Comfort system (Beinshiem, Germany), operating at 5 kVp, 5 mA, 0.16 mm isotropic voxel size, and 15 × 15 cm field of view. Galileos Viewer version 1.8 3D software (Beinshiem, Germany) was employed for evaluation of the CBCT scans.

The evaluation method for the CBCT scans comprised three individual plane assessments (axial, coronal, and sagittal) of root-canal-treated premolar teeth. The following data were recorded for each root-canal-treated premolar:

1. Primary outcome: The presence of different root canal configurations (RCC) in root-canal-treated premolar teeth according to Vertucci’s classification for RCC [31].

2. Secondary outcome: The presence and association of endodontic technical errors and periapical lesions with different RCC.

The obturation quality was assessed according to the criteria stated by Nascimento et al. (2018), with treatment deemed “adequate” when the canal obturation was within 2 mm of the radiographic apex, homogenous in appearance, had no complications or technical errors, and coronal restoration was present. Conversely, quality of treatment was judged “inadequate” when one or more of the preceding technical errors were noted on the CBCT scans:Underfilling, obturation > 2mm short of the radiographic root apex.Overfilling, obturation extruded beyond the radiographic root apex.Non-homogenous obturation, canal obturation with apparent voids.Non-filled canals.Separated endodontic instruments in root canals.Deviation from the root canal’s anatomical path.

In instances of more than one technical error present in a single canal, all errors were documented in combination. Moreover, in cases where the RCC involved multiple root canals, every canal was assessed individually and the presence or absence of technical errors was separately documented for each (Figure 1). Next, the presence or partial or complete absence of coronal restoration was documented and its association with the presence or absence of periapical radiolucency noted (Figure 2) [10]. 

The assessment of periapical status was carried out individually for each root, and apical lesions were deemed present when either a well-defined radiolucency with diameter of >0.5 mm or twice the width of periodontal ligament space was noted around the root tip in more than one view. Apical radiolucency was considered present when a well-defined radiolucent image at least 0.5 mm in diameter or twice the width of the periodontal ligament space was located around the root apex, observed in more than one of the multi-planar views [10,32]. If doubts existed regarding the canal configuration and/or presence or absence of technical errors and periapical lesions, the evaluators were directed to consult the research supervisor (M.Q.J) to reach consensus.

### 2.3. Evaluators’ Intra- and Inter-Reliability and Statistical Analysis

Statistical analysis of data was conducted utilizing SPSS software (version 24.0, SPSS, Inc., Chicago, IL, USA). Data were reported as percentages and frequencies. The Kappa test was applied to calculate the intra- and inter-rater agreement [30], with the significance level set at 5% (α = 0.05). For the primary outcome, intra-rater reliability was determined by comparison of two evaluation scores for the same dataset. Two weeks after the conclusion of first assessment, 25% of the samples were re-evaluated under similar conditions to confirm the method’s reproducibility. The inter-rater reliability was determined prior to data collection during the calibration process. To assess inter-rater reliability for the primary outcome, 16 root-canal-treated premolars from 12 CBCT images were assessed. The 12 CBCT images used for assessing the inter-rater reliability were not included in the study.

## 3. Results

The intra-rater agreement was 0.893 and inter-rater agreement was 0.854, representing excellent agreement according to Ciccheti [33]. The numbers of mandibular and maxillary premolars constituting the sample are presented in Table 1. It was found that 63.10% of the maxillary first premolars had two roots, whereas 5.36% of the maxillary second premolars had two roots. Meanwhile, none of the mandibular premolars had two roots.

Table 2 depicts the distribution of distinct RCC and the presence of different technical errors in each RCC subtype. The most frequently noted RCC in the current study was Type I (199 cases; 77.73%), followed by Type II (26 cases; 10.15%), Type IV (22 cases; 8.59%), Type V (4 cases; 1.56%), Type III (4 cases; 1.56%), and Type VI (1 case; 0.39%). Types VII and VIII were not found in the current study sample. Under-filling and non-homogeneous filling were the technical errors most frequently observed, collectively accounting for 85.9% of the total errors. Vertucci’s Type I RCC was found to have the least technical errors, at 37.7%. A sequential increase in the percentage of technical endodontic errors with increasing RCC complexity was noted, reaching 100 percent with Type VI RCC.

Table 3 outlines the association of technical errors with periapical status in relation to distinct Vertucci’s RCCs. It was noted that in the presence of technical error the prevalence of apical lesions was about 81%.

Table 4 reveals the effect of the presence of coronal restoration on periapical status. It was noted that even with acceptable root canal treatment, the absence of coronal restoration caused apical lesions in 93% of cases. On the other hand, 20% of root canal treatments with technical errors and a good coronal seal survived with apical lesions.

## 4. Discussion

In the present study, all premolars were included, maxillary or mandibular. It was observed that none of the mandibular premolar teeth had more than one root. However, 63% of maxillary first premolars and 5.3% of maxillary second premolars had two roots. These findings are consistent with other studies conducted in Saudi and Pakistani populations to evaluate the root canal anatomy of premolars. These previous studies also reported that most maxillary first premolars possess two roots, whereas the majority of maxillary second premolars and mandibular premolars have single roots [34,35]. A study comparing numbers of roots and root canal configurations between Asian and White ethnic groups reported contradictory results, indicating that in the Asian group 83.2% of maxillary first premolars exhibited a single root, while in the White group, the prevalence of single-rooted maxillary first premolars was 48.7% [36]. 

According to Vertucci’s classification, the most frequently noted root canal configurations in the current study were Type I (77.73%), followed by Type II (10.15%) and Type IV (8.59%). A similar study conducted by Nascimento et al. in 2019 in a Brazilian population reported that the most prevalent RCC in premolar teeth was Type I (71%) and that Type IV occurred more frequently (15%) than Type II (8%) [28]. In a study conducted in Turkey, it was demonstrated that RCC Type IV (76.9%) was the most common for maxillary first premolars (76.9%) and Type I (54.5%) for maxillary second premolars (54.5%). The most prevalent RCC for mandibular first and second premolars was Type I (93.5% and 98.5%, respectively) [37]. The literature provides ample evidence that the root canal anatomy of maxillary and mandibular premolars varies considerably amongst individuals of different ethnicities, ages, and genders [5,6,7,8,14,36].

Several factors can result in endodontic technical errors. These include but are not limited to lack of skill and expertise of the practicing clinician, poor pulp canal access and visibility, inadequate understanding of internal root canal morphology, and wrong choice of imaging modality [38]. One important factor that is often neglected is the complexity of root canal configuration. In our study, a sequential increase in the frequency of technical errors was observed with the increasing complexity of the root canal anatomy. This increase in the number of endodontic errors also led to an increase in the presence of associated periapical lesions. Prevalence of periapical radiolucency was 81% in the presence of technical errors. The study conducted by Nascimento et al. reported similar findings [10,28]. This may be attributed to root canal irregularities, isthmuses, deltas, and ramifications present in the more complex RCC types. These areas of the root canal system are usually inaccessible to mechanical instrumentation and may harbor intracanal bacteria due to incomplete debridement. These pathogenic bacteria can cause apical periodontitis if they gain access to the periapical tissues [39,40,41]. 

In our study, underfilling and non-homogenous filling were the most frequently observed technical errors. One previous study reported similar findings, with underfilling being the most prevalent error present in the sample, but that study was not exclusive to endodontically treated premolars and also included anterior and molar teeth [10]. Nascimento et al. also reported that underfilling of root canals was the most common endodontic technical error in premolars, except in those with RCC Types IV and V or others in which the most frequent kind of error was missed or unfilled canals. [28]. These results differed from those of the current study, in which it was found that 27.2% of premolars with RCC Type IV had received non-homogenous filling and only 4.5% had unfilled canals. Types V and VI displayed unfilled canals as the most common error. A study conducted by Baruwa et al. reported that endodontically treated teeth with non-filled canals were associated with periapical pathology in 82.6% of cases. It can be inferred from these findings that unfilled or missed canals in root-canal-treated teeth can negatively impact prognosis. The study was not limited to premolars and included all endodontically treated anterior and posterior teeth, with maxillary first molars exhibiting the highest proportion of missed canals (59.5%). The prevalence of missed canals in maxillary and mandibular second premolars was reported to be 2.4% and 1.9%, respectively. However, that study did not elaborate on Vertucci’s root canal configurations, their complexities, and their association with missed canals in premolar teeth [11].

In agreement with the present study, Nascimento et al. also reported that endodontically treated premolars with RCC Types I or IV had the fewest endodontic errors [28], clearly due to the less complex configuration of the pulp chamber and canals. RCC Types I and IV lack isthmuses and ramifications, thereby allowing adequate cleaning, shaping, and obturation of canals. Other RCC types are more prone to errors due to multiple canal convergences and ramifications that may be missed on routine periapical radiographs and remain untreated during endodontic therapy. In some RCCs, such as Type II or III, two or more canals merge and exit the tooth through a single foramen. In such cases, adequate sealing of one canal may prevent bacteria from reaching the apex and causing infection despite the presence of a technical error in the other canal. In others, such as RCC Types IV and VIII, canals have multiple apical foramina, and all the foramina must be sealed properly to achieve endodontic success. Leakage through any of the apical or lateral foramina can lead to the development of periapical lesions [42]. This is supported by the results of the current study, in which 75 out of 106 endodontically treated premolars with endodontic technical errors were associated with periapical lesions. 

The current study has demonstrated that even with good endodontic treatment, absence of coronal restoration led to apical lesions in 93% of cases, consistent with the results obtained by Nascimento et al. [28]. The reason for this is that an impermeable coronal seal is crucial for the success of endodontic treatment, preventing the ingress of bacteria into the canals and subsequently into the periradicular area. [43,44,45]. Absence of good coronal restoration and hence a good coronal seal is associated with the development of periapical lesions [32,46]. 

This is the first study conducted in a Saudi population to determine an association between root canal complexities and endodontic treatment errors. The observational design of this study provides the advantage of being less expensive than longitudinal studies while involving a lower risk of bias. One of the limitations of the current study was the size of the voxel used (0.16 mm). Smaller voxel sizes are now available that are more suited for endodontic evaluation of teeth. CBCT may also show widening of the periodontal space in a normal tooth, leading to misdiagnosis of apical periodontitis [47]. In cross-sectional observational studies, data is collected simultaneously allowing no comparison between preoperative and postoperative CBCT images, therefore it cannot be determined whether a periapical lesion had healed, reduced, or enlarged. Therefore, widening of the periodontal space or the presence of periapical radiolucency alone is not conclusive of endodontic failure. Correlation of radiographic findings with clinical signs and patient symptoms is necessary to reach a final verdict. Another factor lacking in this study is the histological component that can establish an association between treatment errors and periapical pathoses. Unrestorable teeth can be evaluated after extraction using scanning electron microscopy to better understand this association [48]. Although adequate, the sample size for this study was limited and further research involving a larger sample will be beneficial. The results of the present study cannot be generalized because the sample was obtained only from a single center and a single population. Further research involving longitudinal study design and data acquired from multiple centers and different ethnic populations is recommended to better understand the association between root canal complexities, endodontic treatment errors, and periapical lesions.

The results of this study highlight the role of pre-operative assessment of root canal configuration for the success of endodontic treatment. A comprehensive understanding of the internal anatomy of a tooth requiring endodontic therapy can not only aid treatment planning but can also contribute to successful treatment without iatrogenic technical errors.

## 5. Conclusions

The frequency of endodontic technical errors increased as root canal configurations became more complex. The most common technical errors observed in endodontically treated premolars were underfilling and non-homogenous filling. There was also an increase in the prevalence of periapical lesions in teeth with endodontic technical errors and/or absence of coronal restoration.

## Figures and Tables

**Figure 1 ijerph-20-01142-f001:**
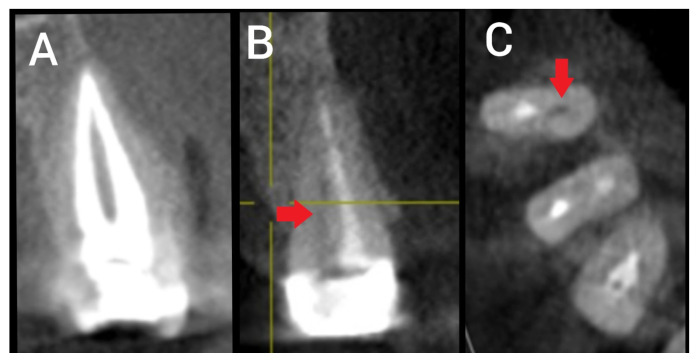
Sections of CBCT showing root canal treatment of two premolars with Type II canal configuration: (**A**) Sagittal section showing root canal treatment of a type II premolar without technical endodontic error; (**B**) sagittal section showing defective root canal treatment of a Type II premolar with a missed untreated canal (red arrow); and (**C**) cross-sectional view of the same premolar tooth showing missed untreated canal (red arrow).

**Figure 2 ijerph-20-01142-f002:**
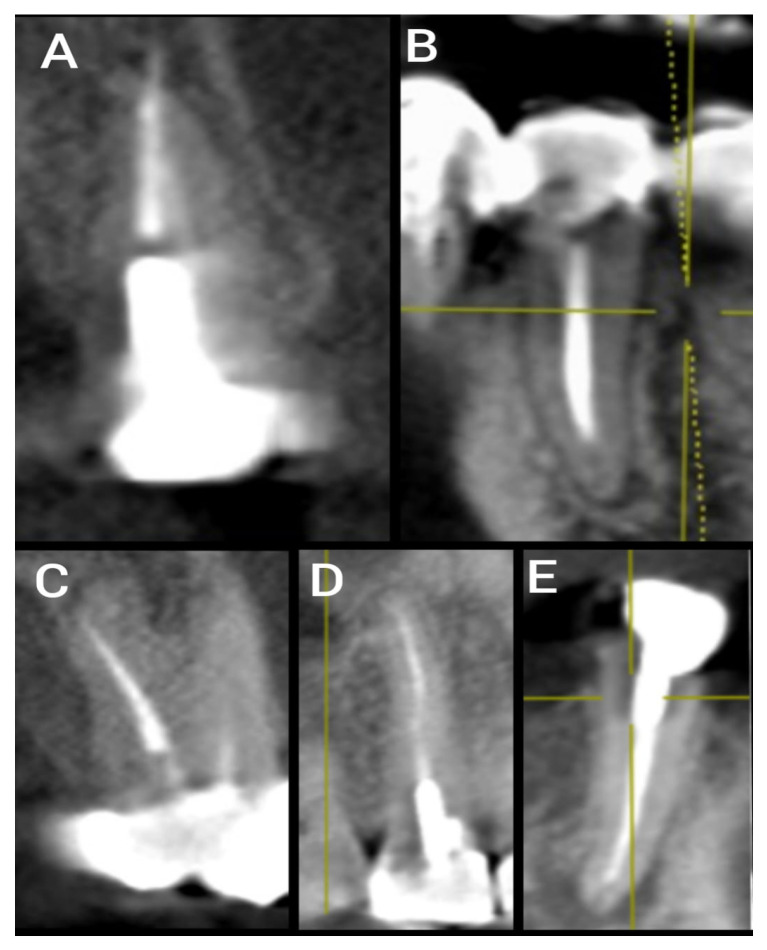
Sections of CBCT showing cases of different RCC of premolar teeth with different technical endodontic errors: (**A**) Sagittal section of treated premolar tooth with extruded overfilled material; (**B**) coronal section of treated premolar tooth with short root canal obturation; (**C**) sagittal section of treated premolar tooth showing deviation from the original path; (**D**) coronal section of treated premolar tooth showing non-homogeneous root canal obturation; (**E**) coronal section of treated premolar tooth with defective coronal seal.

**Table 1 ijerph-20-01142-t001:** Premolar Type * Number of Roots cross tabulation.

Premolars	Roots Number and Percentage		
	One Root	Double-Rooted	Total
First Maxillary (n = 65)	24 (36.9)	41 (63.1)	106 (41.5)
Second Maxillary (n = 56)	53 (94.6)	3 (5.4)	59 (23.0)
First Mandibular (n = 32)	32 (100)	0 (0)	32 (12.5)
Second Mandibular (n = 59)	59 (100)	0 (0)	59 (23.0)
Total (n = 212)	168 (79.2)	44 (20.8)	256

**Table 2 ijerph-20-01142-t002:** Vertucci’s Classification * Error Type cross tabulation.

Vertucci’s Classification * Error Type Cross Tabulation								
Vertucci’s Classification of Root Canal Configuration	Error Type (Frequency and Percentage)							* Total
	No Technical Endodontic Errors	Underfilling	Unfilled Canal	Non-Homogeneous Filling	Deviation	Overfilling	Association of Technical Errors	
Type I (n = 199)	124 (62.3)	45 (22.6)	2 (1)	33 (16.5)	1 (0.5)	4 (2)	10 (5)	75 (37.6)
Type II (n = 26)	13 (50)	5 (19.2)	4 (15.3)	3 (11.5)	0 (0)	1 (3.8)	0 (0)	13 (50)
Type III (n = 4)	2 (50)	1 (25)	1 (25)	0 (0)	0 (0)	0 (0)	0 (0)	2 (50)
Type IV (n = 22)	10 (45.4)	5 (22.7)	1 (4.5)	11 (50)	0 (0)	0 (0)	5 (22.7)	12 (54.5)
Type V (n = 4)	1 (25)	0 (0)	2 (50)	1 (25)	0 (0)	0 (0)	0 (0)	3 (75)
Type VI (n = 1)	0 (0)	0 (0)	1 (100)	0 (0)	0 (0)	0 (0)	0 (0)	1 (100)
Total (n = 256)	150 (58.5)	56 (21.8)	11 (4.2)	48 (18.7)	1 (0.4)	5 (1.9)	15 (5.8)	106 (41.4)

* Total number of roots with at least one endodontic technical error. Coronal restoration status was not considered on this analysis.

**Table 3 ijerph-20-01142-t003:** Apical Radiolucency * Vertucci’s Classification * Technical Error cross tabulation depicting frequency and percentage.

Technical Error			Vertucci’s Classification						Total
			Type I (1-1)	Type II (2-1)	Type III (1-2-1)	Type IV (2-2)	Type V (1-2)	Type VI (2-1-2)	
Present	Apical Radioulency	Present	58 (77.3)	10 (76.9)	2 (100)	12 (100)	2 (66)	1 (100)	85 (80.1)
		Absent	17 (22.6)	2 (23)	0	0	1 (33)	0	20 (19.9)
	Total		75	13	2	12	3	1	106
Absent	Apical Radioulency	Present	26 (20.9)	6 (46.1)	1 (50)	5 (50)	1 (100)	0	39 (26)
		Absent	98 (79)	8 (53.8	1 (50)	5 (50)	0	0	112 (74)
	Total		124	13	2	10	1	0	150
Total	Apical Radioulency	Present	84 (42.2)	16 (61.5)	3 (75)	17 (77.2)	3 (75)	1 (100)	124 (48.4)
		Absent	115 (57.7)	10 (38.4)	1 (25)	5 (22.7)	1 (25)	0	132 (51.5)
	Total		199	26	4	22	4	1	256

**Table 4 ijerph-20-01142-t004:** Apical Radiolucency * Coronal Restoration * Technical Error cross tabulation Depicting Frequency and Percentage.

Technical Error			Coronal Restoration		
			Absent	Present	Total
Present	Apical Radioulency	Present	4 (100)	82 (80.3)	86 (81.1)
		Absent	0	20 (19.6)	20 (18.8)
	Total		4	102	106
Absent	Apical Radioulency	Present	14 (93.3)	24 (17.7)	38 (25.3)
		Absent	1 (6.6)	111 (82.2)	112 (74.6)
	Total		15	135	150
Total	Apical Radioulency	Present	18 (94.7)	106 (44.7)	124 (48.4)
		Absent	1 (5.2)	131 (55.2)	132 (51.5)
	Total		19	237	256

## Data Availability

Data will be made available upon request.

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
