# Peer review of "Root Canal Configuration and Its Relationship with Endodontic Technical Errors and Periapical Status in Premolar Teeth of a Saudi Sub-Population: A Cross-Sectional Observational CBCT Study"

_ijerph, 2023, doi:10.3390/ijerph20021142_

Round 1

Reviewer 1 Report

Thank you for submitting this study to our journal. Here go a few concerns and suggestions:

  • I recommend the author to place the keywords by alphabetic order
  • I suggest to add the study design (cross-section observation study) to the title
  • The introduction is good and provides a wide specter info to the study. However I suggest the author to debate the study DOI 10.1016/j.joen.2019.10.007 which has the prevalence of missed canals in the treatments of the premolars. 
  • What was the ethnic group of the patients?
  • The title mentions Saudi sub-population. Were all patients Saudi Arabian?
  • How were the patients selected? Random? By a specific order? All in?
  • Did the authors conducted inter-rated reliability tests?
  • What visualization software was used?
  • Mentioning that the study has followed a specific checklist will increase the reliability of the study. I recommend the authors to state and adjust the manuscript to the “Preferred Reporting Items for Epidemiologic Cross-sectional studies on Root and Root Canal Anatomy using Cone-Beam Computed Tomographic Technology” and mentioning the study DOI 10.1016/j.joen.2020.03.020
  • May the authors provide the standard deviation on the prevalences in the results?
  • I suggest the authors to debate the indications of using CBCT according to the AAE
  • Please debate the study strength, generalization of the results and further studies recommendations. 
  • Please notice the references are not according to the journal guidelines. 

Reviewer 2 Report

The article is complete and accurate in all details.

I would suggest adding the following work to the bibliography:

Chieruzzi M, Pagano S, De Carolis C; et al. Scanning Electron Microscopy Evaluation of Dental Root Resorption Associated With Granuloma Microscopy and Microanalysis 2015 ; 21 (5): 1264 - 1270 DOI 10.1017/S1431927615014713

Round 2

Reviewer 1 Report

Dear author, I have no more concerns.